# Downregulation of Glycine N-Acyltransferase in Kidney Renal Clear Cell Carcinoma: A Bioinformatic-Based Screening

**DOI:** 10.3390/diagnostics13233505

**Published:** 2023-11-22

**Authors:** Juan P. Muñoz, Gloria M. Calaf

**Affiliations:** 1Laboratorio de Bioquímica, Departamento de Química, Facultad de Ciencias, Universidad de Tarapacá, Arica 1000007, Chile; 2Instituto de Alta Investigación, Universidad de Tarapacá, Arica 1000000, Chile

**Keywords:** clear cell renal cell carcinoma, KIRC, glycine N-acyltransferase, GLYAT, cancer, diagnosis

## Abstract

Clear cell renal cell carcinoma (KIRC) is the most common subtype of renal cell carcinoma (RCC). This form of cancer is characterized by resistance to traditional therapies and an increased likelihood of metastasis. A major factor contributing to the pathogenesis of KIRC is the alteration of metabolic pathways. As kidney cancer is increasingly considered a metabolic disease, there is a growing need to understand the enzymes involved in the regulation of metabolism in tumorigenic cells. In this context, our research focused on glycine N-acyltransferase (GLYAT), an enzyme known to play a role in various metabolic diseases and cancer. Here, through a bioinformatic analysis of public databases, we performed a characterization of GLYAT expression levels in KIRC cases. Our goal is to evaluate whether GLYAT could serve as a compelling candidate for an in-depth study, given its pivotal role in metabolic regulation and previously established links to other malignancies. The analysis showed a marked decrease in GLYAT expression in all stages and grades of KIRC, regardless of mutation rates, suggesting an alternative mechanism of regulation along the tumor development. Additionally, we observed a hypomethylation in the GLYAT promoter region and a negative correlation between the expression of the GLYAT and the levels of cancer-associated fibroblasts. Finally, the data show a correlation between higher levels of GLYAT expression and better patient prognosis. In conclusion, this article underscores the potential of GLYAT as a diagnostic and prognostic marker in KIRC.

## 1. Introduction

Kidney cancer is a relatively rare neoplasm, accounting for about 2.2% of all cancer diagnoses worldwide. It is the 16th most common cancer and was the estimated cause of 431,288 deaths in the year 2020 [1]. Renal cell carcinoma (RCC) is the most common type of kidney cancer, accounting for about 85% of all cases [2]. RCC presents several histological and molecular subtypes, each with its unique characteristics and prognosis. Notably, kidney renal clear cell carcinoma (KIRC), or clear cell renal cell carcinoma (ccRCC), is the most common subtype of RCC and accounts for approximately 80–90% of all cases [3]. KIRC arises from the proximal tubular epithelial cells and is usually recognized by its distinctive histological features and clinical implications [4]. In addition, it often shows resistance to traditional therapeutic regimens and harbors a high likelihood of metastasis [5,6]. Although the precise origin of KIRC remains poorly understood, several factors have been recognized to increase individual susceptibility, including advanced age, obesity, hypertension, smoking, chronic kidney disease, polycystic kidney disease, and kidney stones [7,8].

Recent evidence suggests that metabolic reprogramming plays a fundamental role in KIRC oncogenesis [9]. One of the most important metabolic changes that occurs in KIRC is the shift from aerobic glycolysis to anaerobic glycolysis, a process known as the Warburg effect [10,11]. This shift allows KIRC cells to grow and proliferate even in the absence of oxygen, which makes them more resistant to traditional therapies [11]. Other metabolic changes that have been observed in KIRC cells include an increased glucose uptake, reduction in the expression of Krebs cycle genes, elevation of pentose phosphate pathway genes, increased fatty acid oxidation, and dysregulated amino acid metabolism [4,12,13]. Thus, extensive metabolic reconfiguration affecting glucose, lipid, and amino acid metabolism contributes significantly to the manifestation of the clear cell phenotype. These changes have led to the consideration of kidney cancer as a metabolic disease [14,15].

Glycine N-acyltransferase (GLYAT) is an enzyme that has recently attracted interest in the pathophysiological mechanisms of metabolic diseases. GLYAT orchestrates a crucial biochemical reaction, catalyzing the transfer of acyl groups onto the N-termini of glycine and glutamine. Through this enzymatic activity, GLYAT orchestrates the synthesis of a diverse array of N-acylglycines, thereby facilitating the conjugation of a multitude of substrates [16]. The implications of this process are far-reaching, as it confers the capacity to modulate the chemical properties of these substrates, endowing them with enhanced solubility and facilitating their excretion. This, in turn, holds significant implications for the balance of metabolic homeostasis. 

A key role of GLYAT lies in its participation in the detoxification machinery, operating at the interface of endogenous and exogenous compound metabolism. This pivotal role is reached by catalyzing a vital reaction that involves the interaction between acyl-CoA compounds and glycine. The outcome of this reaction is the formation of Coenzyme A and N-acylglycine, a transformation that renders these compounds more amenable to elimination from the system [17]. This process holds particular significance in the metabolism and detoxification of a spectrum of compounds, encompassing both xenobiotics, such as benzoic acid or salicylic acid; and endogenous organic acids, like isovaleric acid [15]. In accordance with its detoxification capacity, GLYAT effectively helps protect cells from the harmful effects of a wide range of potentially harmful molecules.

At the cellular level, GLYAT is primarily located in the mitochondria, specifically in the inner mitochondrial matrix, due to the presence of a mitochondrial targeting sequence [18]. Although GLYAT is primarily expressed in the liver and kidney [19,20], its altered expression level in other organs, including the breasts, opens up the possibility of its involvement in several diseases and malignancies [15,21,22,23]. In fact, GLYAT downregulation has been observed in various cancer types, suggesting a potential tumor suppressor role [22,24,25,26]. However, the precise impact of GLYAT downregulation on tumorigenic processes remains unclear. More specifically, the consequences of GLYAT downregulation in KIRC are yet to be comprehensively explored.

Given that metabolic pathway aberrations are a hallmark of cancer, and considering the observed downregulation of GLYAT in other malignancies, our study focuses on assessing GLYAT expression levels in KIRC cases. Our goal is to evaluate whether GLYAT could serve as a promising candidate for further investigation in KIRC. Using data from multiple publicly available datasets, we postulate that GLYAT downregulation is a common feature of metabolic cancers, particularly in KIRC. Decreased GLYAT activity may lead to metabolic changes that promote tumor growth and KIRC progression. Thus, this article underscores the potential of GLYAT as a diagnostic and prognostic marker in KIRC.

## 2. Methods

### 2.1. Expression Level Analysis

In this study, the TIMER2.0 [27,28] (http://timer.cistrome.org/, accessed on 25 July 2023) and Sangerbox3.0 [29] (http://vip.sangerbox.com/home.html, accessed on 27 July 2023) resources were utilized to explore the GLYAT mRNA expression across various cancers and their respective normal tissues. This exploration utilized original datasets from The Cancer Genome Atlas (TCGA) and The Genotype-Tissue Expression (GTEx) projects [30]. To analyze protein expression levels, we utilized the University of Alabama at Birmingham Cancer Data Analysis Portal (UALCAN, http://ualcan.path.uab.edu/ accessed on 10 August 2023) [31,32], utilizing data sourced from the Clinical Proteomic Tumor Analysis Consortium (CPTAC). The data from Human Protein Atlas (HPA) [33,34,35] (https://www.proteinatlas.org/ accessed on 16 August 2023) were used to gain insights into the immunohistochemistry of GLYAT in both malignant and healthy tissues. In addition, the quantification of the immunohistochemical images obtained from the HPA database was performed using ImageJ software version 1.53t.

The TNM.com platform (https://tnmplot.com/analysis/, accessed on 16 August 2023) was used for a comparison of gene expression in normal, tumor, and metastatic tissues. This platform uses data from the Gene Expression Omnibus (GEO), GTex, TCGA, and TARGET databases [36].

### 2.2. Mutation Character and Methylation Analysis

The cBioPortal database [37,38,39] (http://www.cbioportal.org/ accessed on 26 August 2023) was applied to analyze the mutation character of GLYAT in pan-cancers. “TCGA Pan Cancer Atlas Studies” was chosen for the cohort, and we entered “GLYAT” in the “Query” module to find the alteration sites, types, and numbers of GLYAT in the “cancer type summary” and “mutation” modules. In the UALCAN database [31,32] (http://ualcan.path.uab.edu accessed on 29 August 2023), the “TCGA gene analysis” function was used to explore the difference in GLYAT DNA promoter methylation levels between tumor tissues and normal tissues. The DNA promoter methylation levels of GLYAT in 8 cancers were obtained.

### 2.3. Immune Cell Infiltration Analysis

The “Immune-Gene” component of the TIMER2.0 online platform [27,28] was employed to investigate the link between GLYAT expression and the presence of immune cells in KIRC. Both immune cells and cancer-associated fibroblasts were chosen for this study. The EPIC [40], TIDE [41], MCP-counter [42], and XCELL [43] algorithms were used to evaluate the infiltration of immune cells. The statistical significance was determined using *p*-values and partial correlation values obtained through the Spearman’s rank correlation test, which was adjusted for tumor purity. The methodologies used by each algorithm for fibroblast quantification are as follows: EPIC was employed to estimate the proportions of fibroblasts within tumor samples. The algorithm implemented quadratic programming to estimate cell proportions that best reconcile with the tumor gene expression data, subject to a summation constraint to unity [40]. MCP-counter provided absolute abundance scores for various immune cell types and stromal populations, including fibroblasts. The approach leverages a curated list of marker genes known to be specific for each cell type. The abundance scores are computed based on the expression levels of these marker genes, assuming a linear relationship with the cell numbers [42]. xCell signatures were utilized to determine cell type enrichment from gene expression data, covering a spectrum of 64 cell types inclusive of fibroblasts. The method integrates a large compendium of gene expression profiles with advanced machine learning algorithms to derive cell-type-specific signatures, employing spillover compensation to account for tissue heterogeneity and closely related cell types [43]. TIDE provides indirect insights into fibroblast infiltration by modeling tumor immune evasion mechanisms that may be influenced by fibroblast activity. TIDE applies gene expression signatures to estimate T-cell dysfunction and exclusion, which are key mechanisms in immune checkpoint blockade resistance [41]. These mechanisms can be affected by the presence and activity of fibroblasts in the tumor microenvironment.

Additionally, single-cell RNA-sequencing (scRNA-seq) data were obtained for an in-depth exploration of the tumor microenvironment (TME) at the single-cell level, utilizing the Tumor Immune Single-Cell Hub version 2 (TISCH2) [44]. The Web-based platform (http://tisch.comp-genomics.org/ accessed on 8 September 2023) provided access to a comprehensive repository of scRNA-seq datasets with detailed cell-type annotations. Single-cell transcriptomic analysis was conducted to assess the expression of GLYAT within individual cells of the TME. The TISCH2 platform enabled the identification of specific cell populations expressing GLYAT and the contextualization of its expression within the heterogeneity of the TME. Then, using the provided annotations in the TISCH2 database, each cell was categorized based on its transcriptomic profile, which allowed for precise cell-type identification. This enabled the correlation of GLYAT expression with specific cell types, including immune and non-immune populations within the TME. Finally, GLYAT expression levels were quantified across individual cells and visualized using the platform’s integrated tools. The expression data were subjected to normalization procedures as per TISCH2’s standardized pipeline to ensure comparability across different cells and datasets. The graphical representations of GLYAT expression across various cell types were generated using TISCH2’s visualization tools. Heatmaps, violin plots, and t-SNE (t-distributed stochastic neighbor embedding) plots were utilized to demonstrate the distribution and relative expression levels of GLYAT within the TME.

### 2.4. Survival Data Analysis

By using the GEPIA2.0 Web tool [45] (http://gepia2.cancer-pku.cn/#index accessed on 9 September 2023), we obtained the survival data for the cancer patients with differentially expressed GLYAT, including overall survival (OS) and disease-free survival (DFS). The cutoff low (50%) and cutoff high (50%) were used as the threshold values to split the lowly expressed and highly expressed groups. The statistical differences were assessed by the log-rank test. The GEPEIA2.0 data were compared with the publicly available Kaplan–Meier Plotter [46] and PrognoScan [47] Web tools, which are available at https://kmplot.com/analysis/ (accessed on 12 September 2023) and http://dna00.bio.kyutech.ac.jp/PrognoScan/, (accessed on 15 September 2023) respectively.

### 2.5. Statistical Analysis

The statistical analysis and graphs were designed using GraphPad Prism version 5.0 software (GraphPad Software, Inc., La Jolla, CA, USA). Student’s *t*-test was used to compare the expression of GLYAT between tumor tissues and corresponding normal tissues. Spearman’s rank correlation test was used to analyze the correlations between GLYAT expression and patient survival, as this is a non-parametric test that is appropriate for analyzing correlations between two non-normally distributed variables. The Kaplan–Meier plotter with a log-rank test was used to confirm the functional values of GLYAT on the patient’s prognosis, as this is a non-parametric test that is appropriate for comparing survival curves between two groups. A *p* < 0.05 was considered to indicate a statistically significant difference: * *p* < 0.05, ** *p* < 0.01, *** *p* < 0.001, and **** *p* < 0.0001.

## 3. Results

### 3.1. GLYAT Is Downregulated in KIRC 

To characterize the role of GLYAT in different tissues and its possible association with human tumors, we first evaluated its expression levels in various tissues, pooling data from different public databases. HPA-derived data from both RNA sequencing and immunohistochemical staining show pronounced expression of GLYAT in the liver and kidneys (Figure 1A), specifically in hepatocytes and proximal tubular cells, respectively (Appendix A). Within these organs, GLYAT is co-expressed with a group of 234 genes, all of which are associated with metabolic functions (Figure 1B and Appendix A). Notably, adipose tissue and breasts also showed a detectable expression of GLYAT, albeit at lower levels.

To evaluate the GLYAT expression level in neoplastic tissues, we utilized the TIMER2.0 resource [27,28], contrasting the mRNA expression profiles of GLYAT between tumor specimens and their corresponding normal samples. The available data indicated a notable downregulation of GLYAT in an array of carcinoma types, namely breast cancer (BRCA), cholangiocarcinoma (CHOL), colon adenocarcinoma (COAD), head and neck squamous cell carcinoma (HNSC), kidney chromophobe (KICH), KIRC, kidney renal papillary cell carcinoma (KIRP), liver hepatocellular carcinoma (LIHC), lung adenocarcinoma (LUAD), lung squamous cell carcinoma (LUSC), rectum adenocarcinoma (READ), stomach adenocarcinoma (STAD), and thyroid carcinoma (THCA) (Figure 2A). To validate these results, we turned to the SangerBox3.0 resource [29] to explore GLYAT expression within the TCGA and GTEx datasets. This Web tool corroborated the downregulation of GLYAT in a majority of the cancers highlighted by TIMER2.0 (Figure 2B). In a focused analysis on KIRC, using the GEPIA platform [45], we observed a marked reduction in GLYAT mRNA expression in KIRC samples relative to matched normal tissues, aligning with the data from TIMER2.0 and SangerBox3.0 (Figure 2C).

To corroborate the findings of the transcriptomic data, we examined the protein expression level of GLYAT in 110 KIRC samples and 84 normal samples by using the UALCAN Web tool. This analysis revealed a statistically significant difference in GLYAT protein levels between KIRC samples and their normal counterparts (*p* = 6.6 × 10^−50^; Figure 2D). This finding confirms that the decrease in GLYAT mRNA observed in KIRC cases is also accompanied by a decrease in protein levels. Furthermore, an analysis of GLYAT from mass spectrometry-based proteomic data from CPTAC confirmation/discovery cohorts in 10 pan-cancer subtypes revealed the same pattern of GLYAT reduction in tumor samples compared to normal samples (Figure 2E). This suggests that the downregulation of GLYAT is a common feature in cancer.

The immunohistochemical analysis of GLYAT protein levels in KIRC samples from the HPA repository provided additional evidence. Figure 2F shows robust cytoplasmic staining in both tubular and glomerular cells of normal kidney tissues. In contrast, tissues derived from renal adenocarcinoma cases showed a tenfold reduction in GLYAT protein levels, as indicated by the absence of peroxidase staining (Figure 2G). These findings support the hypothesis that GLYAT expression is attenuated during the oncogenic transformation of kidney tissue. 

Collectively, our analyses reveal that GLYAT displays elevated expression in hepatic and renal tissues, while its expression is downregulated in cancer samples. 

### 3.2. GLYAT Is Downregulated across Tumor Grade and Cancer Stages in KIRC Patients

The correlation between molecular markers and cancer progression is of paramount importance in advancing our understanding of disease dynamics and clinical management. Building upon this, we investigated the dynamics of GLYAT protein levels in different tumor grades and at different KIRC stages, using different public databases. The analysis of TGCA public data using the UALCAN Web tool consistently demonstrated a decrease in GLYAT expression across all tumor grades and cancer stages in comparison to normal tissue. Nevertheless, no statistically significant differences in GLYAT expression were observed when comparing tumor grades and stages (Figure 3A,B). These results were also confirmed by available transcriptome-level datasets assessed in the TNM Web platform [29] (Figure 3C). Here, the data showed a statically significant downregulation of GLYAT in both tumor and metastatic cells. This observed pattern suggests a potential role of GLYAT in the context of disease progression. Taking this into account, we further characterized the GLYAT gene expression in the tumor microenvironment at a single-cell resolution by using the TISCH Web tool [44]. To do this, we performed a single-cell analysis from different datasets (Appendix A). The results from the GSE159115 and GSE111360 datasets revealed that GLYAT expression was predominantly co-localized to stromal cells, as depicted in Appendix A (red arrows). Contrastingly, the GSE171306 dataset showed that malignant cells also displayed a basal GLYAT expression (Appendix A). 

Collectively, these analyses suggest that GLYAT may be a key protein in the transition between normal and carcinogenic clear renal cells. Furthermore, the data show that GLYAT is conserved at basal levels in the tumor microenvironment of KIRC cells.

### 3.3. GLYAT Is Hypomethylated in KIRC Cases

Given the observed decrease in GLYAT expression in samples of KIRC, we hypothesized that the observed decrease in GLYAT expression in KIRC samples was due to genetic mutations in the GLYAT gene. However, mutation data from the cBioPortal Web tool revealed a GLYAT mutation rate of only 0.2% in KIRC cases (Figure 4). This suggests that the decreased expression of GLYAT in KIRC is not attributable to genomic mutations but may instead be mediated by alternative regulatory mechanisms affecting gene expression, such as DNA methylation.

Aberrant DNA methylation is a major epigenetic mechanism that has a significant impact on gene expression, with implications for tumor initiation and progression. Given the observed reduction in GLYAT expression among KIRC samples, independently of gene mutations, we attribute this reduction to potential hypermethylation of the GLYAT gene’s promoter sequence within KIRC cases. To evaluate this hypothesis, we subjected data from the TCGA project to analysis via the UALCAN Web tool. The results of this analysis revealed a noteworthy discrepancy. Specifically, the methylation of the GLYAT promoter was notably diminished in KIRC cases when compared to normal tissue (*p* = 1.62 × 10^−12^) (Figure 5A). This trend persisted consistently across various tumor grades and distinct stages of cancer progression (Figure 5B,C).

To delve deeper into the methylation characteristics of specific CpG islands within the GLYAT gene, we used the MethSurv Web tool [48] and the UCSC browser [49]. The analysis predicted eight CpG islands (probes) within the GLYAT gene in chromosome 11, of which five showed statistically significant hypomethylation levels in KIRC cases (Figure 5D). Then, the methylation status of these CpG islands was analyzed according to their locations within the GLYAT gene. The results showed that most hypomethylated sites fell on 2000 base pairs of the transcription start site (TSS2000) and the 5’ UTR (Figure 5D). At the same time, methylated sites fell mainly in the 3’UTR and coding sequence regions. Thus, these analyses confirm that regulatory regions of the GLYAT gene have a low methylation level.

We further analyzed how GLYAT expression varied in relation to the methylation levels of each of the CpG islands (Figure 5E). Figure 5F shows that the R-value for the aggregation of all CpG islands is −0.14. This indicates an inverse relation between gene expression and CpG methylation in KIRC gene. These results suggest that the upregulation of the DNA methylation level of these CpGs islands may contribute to the downregulation of GLYAT in KIRC. 

Finally, in order to explore the epigenetic landscape of GLYAT across different cancer types, we used the SMART Web tool from data derived from the TCGA project. The GLYAT methylation level analysis showed a consistent trend of hypomethylation in various cancer types relative to normal tissue, suggesting a shared role for this epigenetic modification of the GLYAT gene in tumor cells. In particular, for KIRC, a statistically significant difference was observed (*p* < 0.001) (Figure 6).

In summary, our analysis revealed a consistent trend of GLYAT gene hypomethylation in different tumor types, suggesting a possible conserved role. Particularly in KIRC, this phenomenon was observed across various tumor grades and cancer stages when compared with the normal tissue. Hypomethylation of specific predicted CpG islands, particularly in the promoter and coding regions of GLYAT, suggests that GLYAT downregulation in KIRC is not due to methylation. This is because CpG islands in these regions are typically hypermethylated to silence genes, while hypomethylation generally leads to increased gene expression. Therefore, other transcriptional and post-transcriptional modifications, unrelated to DNA methylation, may be responsible for GLYAT’s downregulation.

### 3.4. miRNA Activity Associated to GLYAT Downregulation in KIRC

To elucidate potential miRNAs contributing to the downregulation of GLYAT expression in KIRC tumors, we conducted a miRNA target prediction analysis, employing the algorithms provided by miRTarbase and miRTarget Link 2.0. The outcomes of these analyses unveiled a collection of 18 miRNAs predicted to target the 3’-UTR of the GLYAT gene (Appendix A). Subsequently, the expression levels of these identified miRNAs were compared between KIRC tumors and corresponding normal tissues, utilizing a t-test statistical approach. Remarkably, the results demonstrated that none of the predicted miRNAs exhibited a statistically significant increase in expression within KIRC tumors when contrasted with matched normal tissues. These findings suggest that the downregulation of GLYAT expression in KIRC tumors could be independent of miRNA regulation.

### 3.5. Association between GLYAT and Tumor-Infiltrating Immune Cells

Kidney cell cancer has long been identified as a tumor type that elicits an immune response, and numerous immune cells within the tumor microenvironment can contribute to combating the tumor [30]. In order to analyze how GLYAT expression is related to immune infiltration in KIRC, we used the TIMER2.0 resource. All the algorithms used in this analysis (EPIC, MCP-counter, EXCELL, and TIDE) showed that a consistent negative correlation exists between the expression of the GLYAT gene and the levels of cancer-associated fibroblasts (CAF) in KIRC (Figure 7A–D). The statistical significance of this negative correlation was confirmed across the board by all algorithms. Specifically, the MCP-COUNTER algorithm reported a correlation coefficient of −0.408, with a low *p*-value of 5.83 × 10^−20^, thus reinforcing the strength and validity of this analysis. It is crucial to underline the potential impact of tumor purity as a confounding factor in analyses that investigate the relationship between gene expression and immune cell infiltration. In this study, we took this into account by selecting the “Purity Adjustment” option available in the TIMER2.0. This feature adjusts the results for the proportion of cancer cells in the sample, thereby refining the correlation statistics between GLYAT expression and CAF levels. The findings of this analysis suggest that GLYAT expression may be a potential biomarker for CAF infiltration in KIRC.

### 3.6. GLYAT Expression on Disease Prognosis

In order to know the relationship between GLYAT expression and disease prognosis, data from the TCGA, GTEx, and GEO databases were assessed using three online resources: GEPIA2.0, Kaplan–Meier Plotter, and PrognoScan. For the analysis, patients were categorized into two groups based on their GLYAT expression levels: those with high expression and those with low expression. The data analysis indicated that, in the KIRC subtype, patients with high GLYAT expression had significantly better overall survival. The hazard ratio (HR) values were 0.48, 0.42, and 0.35 by using GEPIA2.0, Kaplan–Meier Plotter, and PrognoScan, respectively (Figure 8A–C). Additionally, GLYAT expression was also examined in relation to disease-free survival (DFS). The data presented in Figure 8D show that patients with a high GLYAT expression improved DFS (*p* = 0.012) according to GEPIA2.0. These findings underscore the potential of GLYAT expression as a valuable prognostic indicator, shedding light on its impact on both overall survival and disease-free survival in the studied disease context.

## 4. Discussion

Despite the wide array of diagnostic markers and treatment modalities currently available, KIRC continues to present a dire prognosis in its advanced stages, thereby necessitating the identification of novel molecular targets for both diagnosis and therapy. In this context, GLYAT emerges as a compelling candidate for in-depth study, given its pivotal roles in metabolic regulation and previously established links to other malignancies. Specialized in the biochemical conjugation of xenobiotics like benzoic acid with the amino acid glycine [50], GLYAT is more than a mere metabolic workhorse; it also serves as a key modulator of glycine availability, an aspect of singular relevance in cancer physiology. In particular, the role of GLYAT in mitigating glycine absorption or biosynthesis can have downstream implications on cancer cell growth, likely by hindering the synthesis of nucleic acids, thus disrupting a key metabolic dependency of rapidly proliferating cancer cells.

The observed downregulation of GLYAT in KIRC and other cancer types is notable, particularly given its pronounced baseline expression in renal and hepatic tissues. These results demand further mechanistic inquiries into the role of GLYAT in cancer physiology, given that its primary expression sites, the liver and kidneys, are metabolically active organs. Taking into account these data, it is conceivable that GLYAT could be involved in metabolic pathways disrupted during oncogenesis. In fact, Ren et al. (2017) demonstrated that GLYAT downregulation suppresses cell death and JNK pathway activation [51]. It is therefore plausible that GLYAT could contribute to metabolic reprogramming during cell transformation. 

On the other hand, the absence of genetic mutations in the GLYAT gene in KIRC cases directs our attention to other modes of regulation, particularly epigenetic mechanisms. While the hypomethylation of GLYAT was observed, the moderate correlation coefficient with its expression suggests that methylation is likely only one part of a more complex regulatory network. It would be important to investigate other epigenetic mechanisms, like histone modifications or chromatin remodeling. Moreover, the lack of an observed impact from miRNA expression on GLYAT downregulation adds another layer of complexity. It is conceivable that post-transcriptional mechanisms other than miRNA could be at play, such as translational inhibition or increased protein degradation rates, and these warrant future explorations.

The consistency in the downregulation of GLYAT across different tumor grades and stages in KIRC without a relation to differentiation raises significant clinical questions. Could GLYAT be an early marker for KIRC onset? Furthermore, given that GLYAT expression was primarily co-localized to stromal cells in the tumor microenvironment, one must consider the possible interplay between GLYAT and stromal cells in cancer progression. It would be worth investigating whether GLYAT has a role in stromal-to-tumor signaling pathways, potentially impacting cancer cell metabolism, growth, or invasion.

The negative correlation between GLYAT expression and levels of cancer-associated fibroblasts (CAFs) invites speculation as to whether GLYAT may have an inhibitory role on CAFs or is downregulated in response to signals from CAFs. This correlation, given its significance, could also be an avenue for therapeutic targeting, which would require functional studies to confirm the causality and directionality of this relationship.

The observed positive impact of high GLYAT expression on overall and disease-free survival underscores its potential as a prognostic marker. However, these are correlational observations, and more rigorous, longitudinal studies are needed to establish causality. The potential for GLYAT as a therapeutic target should be evaluated in preclinical models to assess whether its upregulation could indeed confer a survival advantage or slow down disease progression.

While prior research has shed light on the importance of GLYAT in malignancies such as breast cancer and hepatocellular carcinoma [22,24,25,26], its role in the pathophysiology of KIRC remains an underexplored frontier. Given its myriad of functions—from xenobiotic detoxification to metabolic regulation—the examination of GLYAT in the context of KIRC offers a rich avenue for the discovery of both novel therapeutic agents and robust diagnostic markers. Delving into the specific contributions of GLYAT to the biology of KIRC may drive underlying mechanistic pathways that contribute to the tumor’s development, aggressiveness, and resistance to current therapies. As such, a detailed investigation into GLYAT’s impact on KIRC could hold the key to unlocking new strategies for the more effective management of this challenging disease.

An important limitation of this study is the lack of functional data on the consequences of GLYAT downregulation in KIRC tumors. Specifically, we do not fully understand the molecular and cellular mechanisms by which GLYAT downregulation contributes to KIRC oncogenesis. To address this gap in knowledge, future studies should prioritize comprehensive biochemical analyses, in vivo studies, and functional assays to elucidate the effects of GLYAT downregulation on cellular proliferation, apoptosis, metabolic reprogramming, and other cellular activities implicated in cancer development and progression. Additionally, these studies should include larger and more detailed clinical information, as well as functional studies to assess the role of GLYAT in tumor cell proliferation, migration, and survival.

In summary, the data shown here suggest that GLYAT appears to play a multifaceted role in cancer, particularly in KIRC. The widespread downregulation of GLYAT in KIRC tumors, regardless of grade or stage, suggests its involvement in early tumor development and as a potential marker of tumor aggressiveness. Additionally, GLYAT may be a promising diagnostic and prognostic marker for KIRC patients. Moreover, given GLYAT’s known role in cell growth and migration, its downregulation in KIRC tumors may contribute to tumor progression. Future studies should investigate GLYAT’s metabolic functions, epigenetic regulation, role in the tumor microenvironment, and impact on disease progression and patient survival to elucidate its precise role in KIRC tumorigenesis and progression.

## Figures and Tables

**Figure 1 diagnostics-13-03505-f001:**
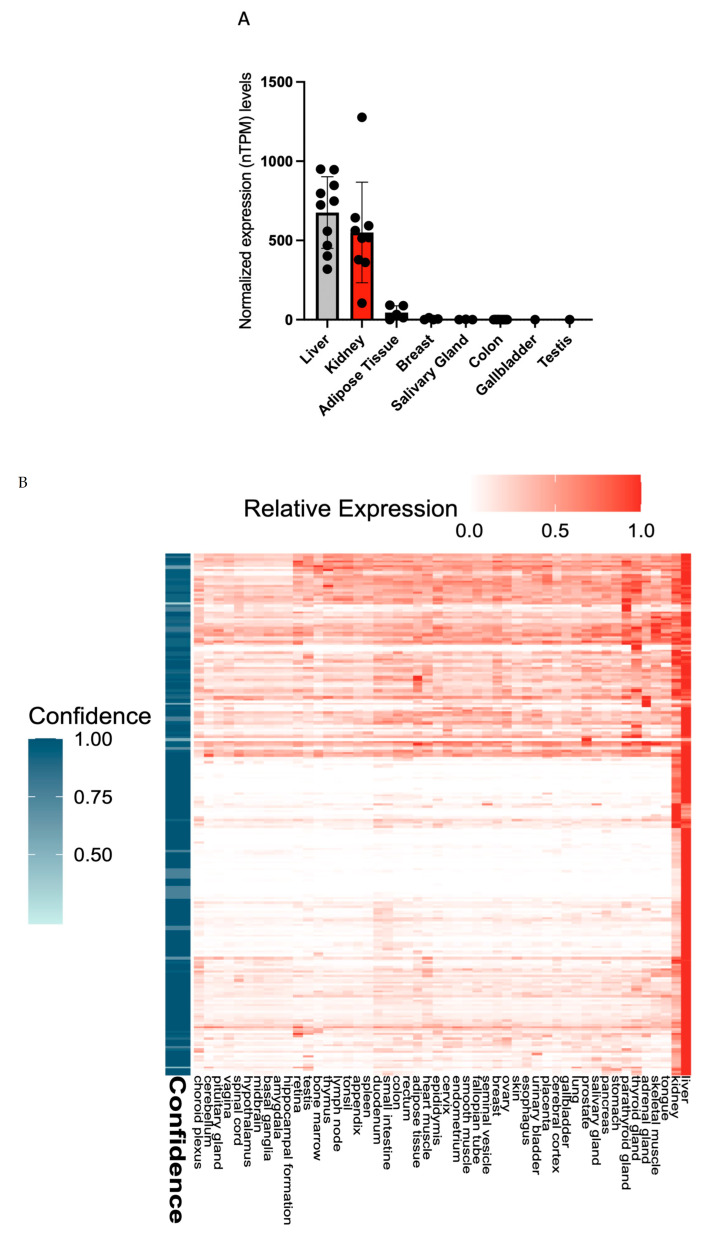
GLYAT is overexpressed in the liver and kidneys. (**A**) Normalized transcripts per million (nTPM) values of GLYAT in different tissues. The graph was generated by merging transcriptomics data from both the human protein atlas (HPA) and the GTEx databases. (**B**) Heat map showing the expression of the 234 genes within the metabolic cluster. Along the horizontal axis, various sample types are delineated, while the vertical axis catalogs the genes attributed to the selected cluster. The color gradient of the heat map indicates gene expression levels, which were normalized to relative expression values. A vertical indicator line accompanies each gene, indicating the confidence level of its assignment to the specific cluster. This confidence score is calculated by evaluating the frequency with which the gene appears in this cluster across multiple iterations and is quantified on a scale from 0 to 1. Data credit: HPA. Images and data (v23.0) are available from proteinatlas.org.

**Figure 2 diagnostics-13-03505-f002:**
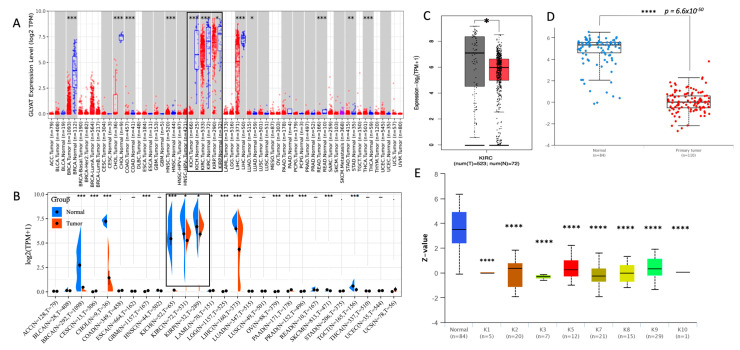
GLYAT expression decreases in tumor samples. Differential expression analysis of GLYAT in TCGA data analyzed via TIMER2.0M (**A**) and SangerBox3.0 (**B**). Data from kidney-related cancers are indicated by black boxes. (**C**) Boxplot of GLYAT mRNA expression in KIRC (red) and normal (gray) tissues analyzed by GEPIA2 from TGAC data. (**D**) GLYAT protein expression comparison between normal (*n* = 84) and primary tumor samples (*n* = 110) obtained from the UALCAN Web tool. Z-values represent standard deviations from the median across samples for the given cancer type. (**E**) GLYAT proteomic expression in pan-cancer subtype obtained from UALCAN. The graph is derived from proteomic data obtained by mass spectrometry from the CPTAC confirmation/Discovery study cohorts of 532 cases categorized into ten different pan-cancer categories (K1 to K10). Z-values represent standard deviations from the median across samples for the given cancer type. (**F**) Immunohistochemical analysis of GLYAT protein expression was conducted by utilizing peroxidase-conjugated antibodies HPA040251 and HPA044094, employing specimens derived from both normal kidney tissue and kidney adenocarcinoma cases. Image credit: Human Protein Atlas. image (v23.0) available from proteinatlas.org. (**G**) The quantification of immunohistochemical staining involved the assessment of six normal kidney samples and seven kidney adenocarcinoma cases. The resulting graph depicts the percentage of stained areas. * *p* < 0.05, ** *p* < 0.01, *** *p* < 0.001, and **** *p* < 0.0001.

**Figure 3 diagnostics-13-03505-f003:**
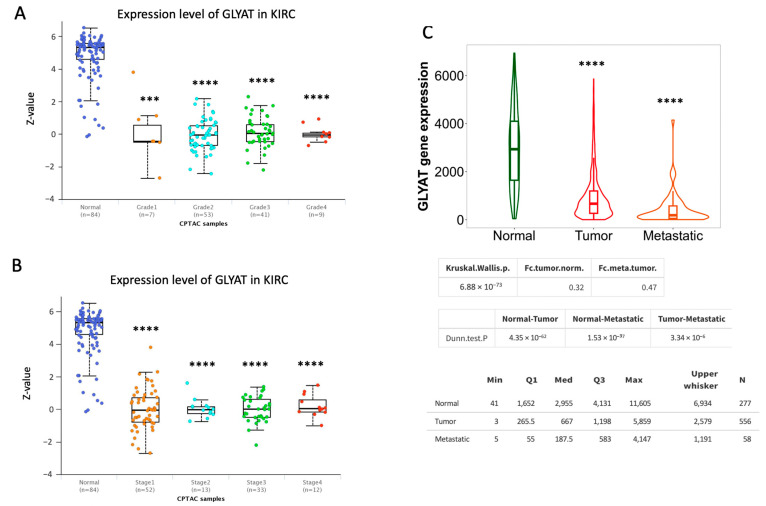
GLYAT expression decreases with tumor grade and cancer stage in KIRC cases. (**A**) Jitter plot of GLYAT protein expression in different tumor grades obtained from the UALCAN. Z-scores represent standard deviations from the median across samples for the given cancer type. (**B**) GLYAT proteomic expression profile based on tumor grade of KIRC obtained from UALCAN resource. (**C**) Violin plot of differential GLYAT expression in normal, tumor, and metastatic cells of KIRC retrieved in the UALCAN. *** *p* < 0.001, and **** *p* < 0.0001.

**Figure 4 diagnostics-13-03505-f004:**
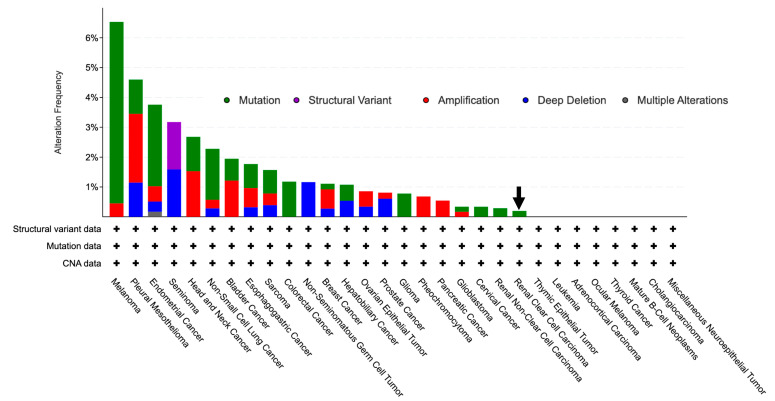
GLYAT alteration frequency in pan-cancers. Barr plot depicting genetic alterations rates of GLYAT in different tumor types from the TCGA database, retrieved from cBioPortal, (https://www.cbioportal.org/, accessed on 22 August 2023)). Mutations are represented in green, deletions in blue, amplification in red, and multiple alterations in gray. Black arrow indicates mutation frequency on KIRC.

**Figure 5 diagnostics-13-03505-f005:**
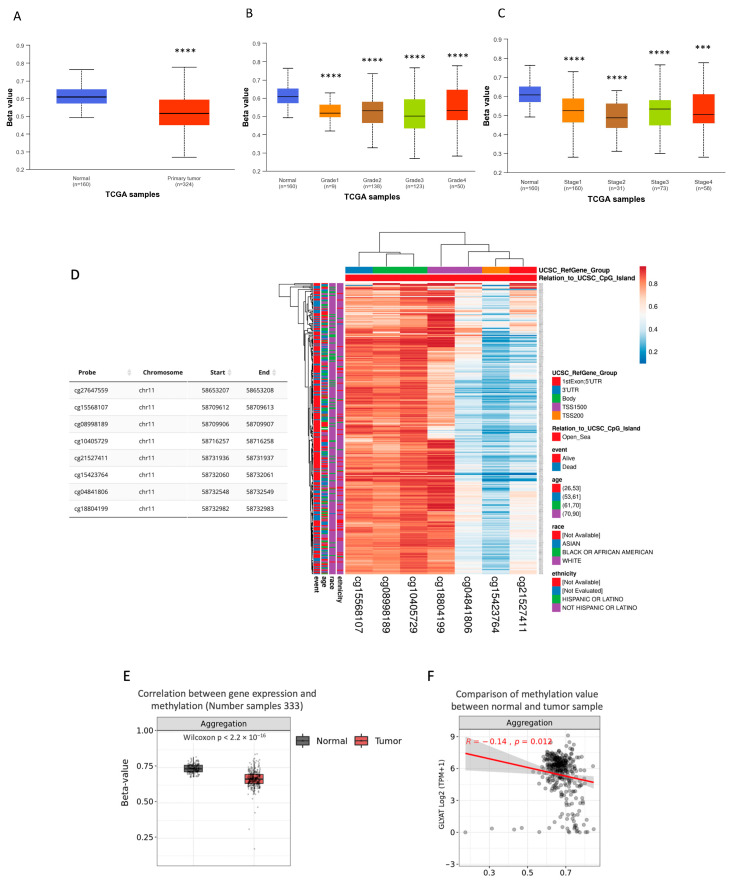
GLYAT promoter is hypomethylated in KIRC cases. (**A**) Data from the TCGA project show the relative DNA methylation in GLYAT promoter based on the beta-value, ranging from 0 (unmethylated) to 1 (fully methylated). (**B**) Methylation levels of the GLYAT promoter in different tumor grades in KIRC cases from the TCGA project. (**C**) Methylation levels of the GLYAT promoter in different cancer stages in KIRC cases. Plots were obtained and modified from the UALCAN Web tool. (**D**) Heat map showing the methylation levels of the GLYAT gene in predicted CpG sites in different genomic regions. The color scale ranges from red (high methylation) to blue (low methylation). The side-boxes show the event, age, race, and ethnicity of the samples. Data were obtained from MethSurv and modified by the ClustVis tool. The visualization of CpG site cg27647559 on the heat map was not available on the database. (**E**) Differential methylation levels of GLYAT probes between normal samples (*n* = 157) and KIRC patients (*n* = 333) from the TCGA project. (**F**) Spearman’s correlation between methylation and mRNA level (Log2-scaled, TPM + 1) of GLYAT in KIRC samples. Data were obtained from TCGA project and analyzed by Methsurv Web tool. *** *p* < 0.001, and **** *p* < 0.0001.

**Figure 6 diagnostics-13-03505-f006:**
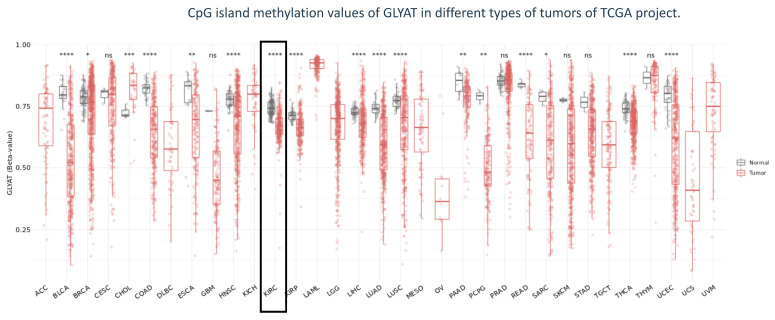
Methylation analysis of CpG islands in TCGA tumors. Jitter plot depicting the beta-value of CpG methylation of GLYAT across different types of tumors from the TCGA project. Black box indicates data of KIRC. The methylation levels are depicted using beta-values, with a scale from 0, indicating unmethylated CpG islands, to 1, representing highly methylated CpG islands. Significance levels are indicated as follows: ns (non-significant): * *p* ≤ 0.05, ** *p* ≤ 0.01, *** *p* ≤ 0.001, and **** *p* ≤ 0.0001.

**Figure 7 diagnostics-13-03505-f007:**
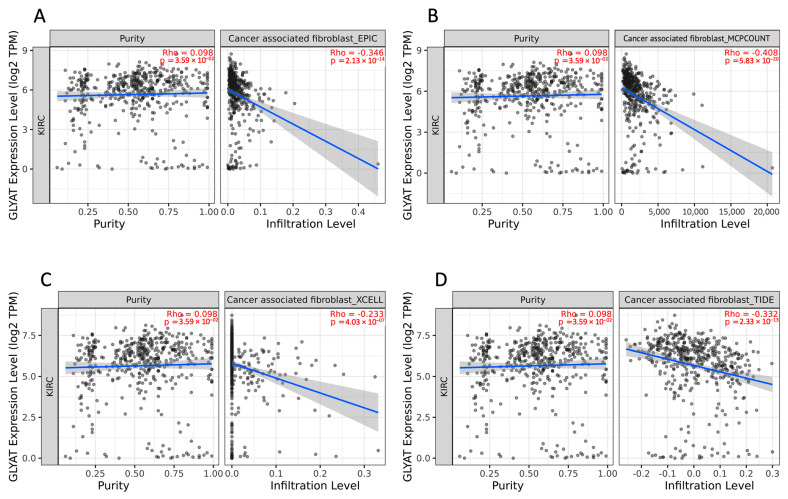
Correlation between GLYAT expression and CAF infiltration in KIRC tumors. Correlation between GLYAT gene expression, represented as Log2 TPM, and cancer-associated fibroblast (CAF) infiltration levels in KIRC tumors quantified by four different algorithms, namely (**A**) EPIC, (**B**) MCPCOUNT, (**C**) CXELL, and (**D**) TIDE, obtained from TIMER2.0 database. The left side of each plot shows the GLYAT expression level against tumor purity. The right side of each plot shows the corresponding CAF infiltration level. The analysis provides insights into potential connections between GLYAT expression and CAF presence within the tumor microenvironment.

**Figure 8 diagnostics-13-03505-f008:**
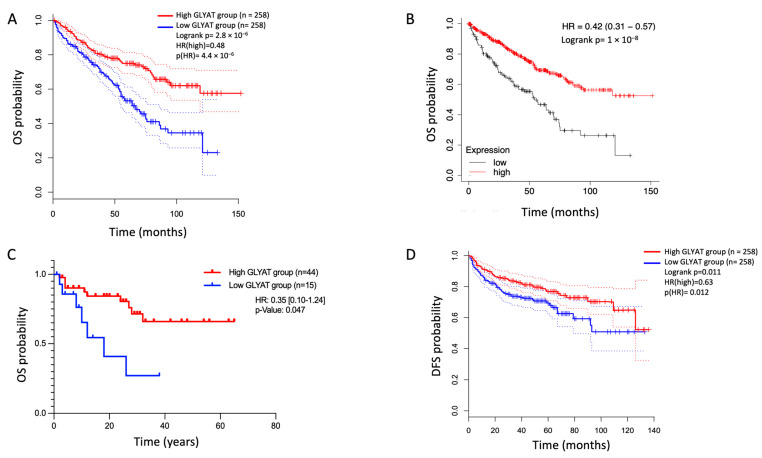
The GLYAT expression on KIRC prognosis. Kaplan–Meier plots depicting the association of GLYAT expression with overall survival (OS) probability and disease-free survival (DFS) in KIRC patients. Patients were categorized into two groups based on their GLYAT expression levels: those with high expression (red line) and those with low expression (blue line). The datasets used in this analysis were sourced from three key online tools: GEPIA2.0 (**A**), the Kaplan–Meier Plotter (**B**), and PrognoScan (**C**). These platforms provided access to primary data derived from the TCGA and GTEx projects, as well as the GEO database. (**D**) Kaplan–Meier plot of DFS according to the GEPIA2.0.

## Data Availability

The results published here are in whole or in part based upon original data generated by the TCGA Research Network, https://www.cancer.gov/tcga; GTEx, https://www.gtexportal.org; Human Protein Atlas, https://www.proteinatlas.org/; CPTAC, https://proteomics.cancer.gov/programs/cptac; and UCSC browser, https://genome.ucsc.edu/.

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
