# Peer review of "Downregulation of Glycine N-Acyltransferase in Kidney Renal Clear Cell Carcinoma: A Bioinformatic-Based Screening"

_diagnostics, 2023, doi:10.3390/diagnostics13233505_

Round 1

Reviewer 1 Report

Comments and Suggestions for Authors

Author Response

In this manuscript, multiple datasets of different types from several sources were analyzed to study the GLYAT expression in Clear cell renal cell carcinoma (KIRC). The GLYAT expression was down regulated in the KIRC compared to normal samples. This was consistent with the down regulated GLYAT protein in the KIRC. This study showed that the GLYAT expression was down regulated across different tumor stages in the KIRC patients, and this might be modulated by epigenetic changes not by mutation and miRNA regulation. In addition to these important findings, they also found that the GLYAT expression and the levels of cancer-associated fibroblasts were negatively correlated, and the GLYAT was useful as a prognostic marker for the KIRC patients. However, the manuscript is not ready for publication in the current form.

Dear Reviewer,

Thank you for your suggestions and comments. We have carefully reviewed and addressed all of them. Below are the details for each point.

The major comments.

  1. P.8, Fig.3A and B, is there any trend from lower grade to higher grade? Is there any difference comparing one grade to another grade?

Regarding the trends in Figs. 3A and 3B, we did not find a statistically significant difference in GLYAT level between tumor grades or tumor stages. Given the main objective of our manuscript, we omitted comparative analysis of GLYAT levels between tumor grades and tumor stages. Additionally, to avoid overcomplicating the visual presentation, we excluded the nonsignificant distinctions between these groups. We believe this decision maintains the graphs' clarity and readability, ensuring that the focus remains on the most pertinent variables. To ensure that readers are fully informed, we included a detailed explanation in the manuscript (lines 258-263), explaining the rationale for the graphical representation and providing the statistical results of the comparisons between the different grades.

In Fig. 5D, the methylation data for cg27647559 is missing. Why?

In response to the inquiry about the missing methylation data in Fig. 5D, it is important to note that the database Methsurv, which was employed for this analysis, did not have the information available for the CpG site cg27647559. This limitation has been acknowledged and clarified in the manuscript on lines 321-322.

  1. P.14, Line 322-323, the authors wrote this “to conclude to the observed downregulation of GLYAT expression in KIRC is independent of this kind of epigenetic regulation”. It is not clear how could the authors reach this conclusion giving that for some CpG sites the methylation and GLYAT expression was negatively correlated.

This conclusion is drawn from the fact that CpG islands within the promoter and coding regions of a gene are typically regions where methylation, often silencing genes when methylated. Hypomethylation in these regions generally leads to an increase in gene expression. However, in the case of KIRC, despite the presence of hypomethylation in these crucial regions, the GLYAT gene is found to be downregulated. This suggests that other regulatory mechanisms may be at play, such as histone modification, miRNA interference, or other transcriptional and post-transcriptional modifications that are not directly related to DNA methylation. Therefore, the regulation of GLYAT in KIRC must be influenced by epigenetic or genetic alterations beyond the methylation status of CpG islands in its promoter and coding regions, challenging the often straight forward association between hypomethylation and increased expression. This explanation was added in the manuscript on the lines 357-362.

  1. Since the GLYAT expression was important for the KIRC patient survival, it is worth studying themethylation difference between GLYAT high group vs. GLYAT low group.

Our analysis of publicly available databases revealed that they can be used to assess the methylation status of various CpG islands in KIRC cases compared to normal controls. We identified distinct methylation profiles that correlated with differential GLYAT expression levels (high in normal controls and low in KIRC), as shown in Figures 5E, 5F, and 6. However, these databases do not currently provide data on the methylation status of GLYAT in patient groups stratified by survival outcomes (i.e., high vs. low GLYAT expression). This limitation persists at the time of writing, and to our knowledge, has not been addressed in the available databases.

The minor comments.

  1. P.5, Line 163, in the caption for Fig.1C, it is better to replace the sentence “Heat map describing gene expression within the metabolic cluster” by “Heat map showing the expression of the 234 gene within the metabolic cluster”.

Thank you for your suggestion. The sentence was changed. please, check the line 195

  1. P.13, Line 280, “betavalue” should be beta-value. In Fig.6 caption, “b-values” should be betavalues.

The sentence has been changed. please, check the line 315.

  1. P.10, Fig. 3G, the figure legend “Celltype (malig” is incomplete.

The figure has been improved. Please, see the new Fig. 3G. Thank you for your thorough analysis.

  1. P.10, Line 245, delete the repeated descripton for (D).

The repeated description has been removed.

  1. In the four KM-plots in Fig. 8, the y-axis should be labeled either by OS or DFS probability.

The KM-plots were modified

  1. The figure legends in Fig.8A and D should be placed at the bottom-left of the plots.

The legends were modified.

We appreciate your valuable comments and time spent reviewing our manuscript. Your suggestions have been invaluable in improving our work.

Reviewer 2 Report

Comments and Suggestions for Authors

In the present manuscript the authors investigated Glycine N-acyltransferase (GLYAT) mainly in clear cell renal cell carcinoma (KIRC). Although there is no formal validation with own tumour samples, very comprehensive bioinformatics analyses were conducted to implicate GLYAT in the tumorigenesis of KIRC. Results are sound and the conclusions are supported by the data. Overall, there are no major issues in the manuscript. Thus, it is appropriate for the journal. Minor comments are as follows:

- The rationale of focusing this study on GLYAT and not another gene should be more clearly indicated.

- Figure 2F shows the immunohistochemical expression of GLYAT in 3 KIRC samples compared to 3 normal kidneys. However, an unbiased quantification of all KIRC samples compared to all normal kidneys should be performed.

- The methodology for quantifying lymphocyte infiltration should be more detailed, indicating the parameters used on each algorithm, as well as the procedure for the single-cell analysis.

Comments on the Quality of English Language

English language quality is appropriate.

Author Response

In the present manuscript the authors investigated Glycine N-acyltransferase (GLYAT) mainly in clear cell renal cell carcinoma (KIRC). Although there is no formal validation with own tumour samples, very comprehensive bioinformatics analyses were conducted to implicate GLYAT in the tumorigenesis of KIRC. Results are sound and the conclusions are supported by the data. Overall, there are no major issues in the manuscript. Thus, it is appropriate for the journal. Minor comments are as follows:

Dear Reviewer, we are grateful for your constructive feedback on our manuscript. We have carefully reviewed and addressed all of your suggestions, and we believe that the manuscript has been significantly improved as a result. Below are the details for each point, along with a brief explanation of how we have addressed your feedback.

- The rationale of focusing this study on GLYAT and not another gene should be more clearly indicated.

An explanation detailing the rationale behind focusing this study specifically on the gene GLYAT, has been included in the introduction of the manuscript. This can be found on lines 83-90, where we have elaborated on the reasons and significance of selecting GLYAT in our research.

- Figure 2F shows the immunohistochemical expression of GLYAT in 3 KIRC samples compared to 3 normal kidneys. However, an unbiased quantification of all KIRC samples compared to all normal kidneys should be performed.

We have added a quantification of the images. See new Figure 2G. We have also added the detail of the methodology to the Materials and methods section and to the figure caption.

- The methodology for quantifying lymphocyte infiltration should be more detailed, indicating the parameters used on each algorithm, as well as the procedure for the single-cell analysis.

The details regarding the methodology for quantifying fibroblast infiltration have been included in the Materials and Methods section. Please review lines 123-159 for a comprehensive explanation of the parameters utilized in each algorithm and the procedure for single-cell analysis.

We are grateful for your constructive comments on our manuscript. Your feedback has helped us to clarify our arguments and improve the overall readability of our work.

Round 2

Reviewer 1 Report

Comments and Suggestions for Authors

Minor comments:

1.       In Figure 8A, B and C, it is better to use “OS probability” to label the y-axes.

2.       The figure legends and the survival curves should not overlap in Fig.8 A and D.

3.       P.15, Line 381, some references are needed for some key algorithms such as EPIC, MCP-COUNTER, EXCELL and TIDE.

4.       In the figure caption for Figure 5 (A), delete “beta-value.” to combine the two sentences into one.

Comments on the Quality of English Language

The Quality of English for this manuscript is good.

Author Response

Dear Reviewer,
Thank you for your suggestions on our manuscript. We have considered your feedback and made all of the recommended changes. Below, we provide a detailed summary of the changes made:

  1. In Figure 8A, B and C, it is better to use “OS probability” to label the y-axes.

The y-axes of Figures 8A, B, and C have been changed to "OS probability," as suggested.

  1. The figure legends and the survival curves should not overlap in Fig.8 A and D.

The figure legends and survival curves in Figures 8 have been modified so that they no longer overlap.

  1. P.15, Line 381, some references are needed for some key algorithms such as EPIC, MCP-COUNTER, EXCELL and TIDE.

References for the algorithms have been added to the Materials and Methods section, along with citations for all resources and databases used in the study

  1. In the figure caption for Figure 5 (A), delete “beta-value.” to combine the two sentences into one.

The figure caption for Figure 5A has been improved by combining the two sentences into one and deleting the phrase "beta-value"